# The Perception of Overweight and Obesity among South African Adults: Implications for Intervention Strategies

**DOI:** 10.3390/ijerph191912335

**Published:** 2022-09-28

**Authors:** Mashudu Manafe, Paul Kiprono Chelule, Sphiwe Madiba

**Affiliations:** 1Department of Human Nutrition & Dietetics, Sefako Makgatho Health Sciences University, Pretoria 0001, South Africa; 2Department of Public Health, Sefako Makgatho Health Sciences University, Pretoria 0001, South Africa; 3Faculty of Health Sciences, University of Limpopo, Mankweng 0727, South Africa

**Keywords:** perceptions, overweight, obesity, beliefs, health, obesity-related conditions

## Abstract

Obesity is a public health problem, affecting more than half of the global population. Perceptions and beliefs play a significant role in preventing and managing overweight and obesity. Thus, the paper explores own perception of overweight and obesity on personal health among South African adults. This was a qualitative study in which the participants (*n* = 24) were interviewed in-depth on their perception of obesity and health. Thematic analysis was used in data assessment. The results revealed four main themes: perceived causes of obesity, self-perception of being obese, perception of obesity, health, and cultural beliefs. Environmental and personal factors were perceived as the leading cause of overweight and obesity. The findings further showed that body weight preference was personal and differed from individual to individual. The study provided insight into perceptions of obesity from individuals’ perspectives. The misperceptions of overweight and obesity are helpful in the design of impactful intervention programmes that can be used to prevent and manage obesity in South Africa.

## 1. Introduction

The rise in obesity poses a challenge to public health worldwide. South Africa is one of the countries with the highest obesity prevalence, with a projected increase in obesity by 47.7% in females and 23.3% in males by 2025 [1]. The higher prevalence of obesity leads to an increase in non-communicable diseases (NCDs) such as type 2 diabetes mellitus (T2DM), cardiovascular diseases, and hypertension which are currently the biggest threats to health and development among populations [2]. A systematic review conducted by Kramer et al. [3] showed that persons with obesity are at increased risk of long-term, adverse health outcomes, even without metabolic comorbidities such as high blood pressure.

The causes of obesity are multifactorial and include environmental and individual factors. Consumption of energy-dense foods is linked to the increased prevalence of overweight and obesity [4]. However, the consumption of obesogenic foods is more likely observed in urban areas where energy-dense foods associated with a western lifestyle have been adopted [5,6]. Cultural perceptions and ideals fuel the increase in overweight and obesity, particularly in Africa. For example, it is commonly believed that healthy people should not be skinny as it symbolises poverty and ill health [7].

Moreover, obesity is often linked to happiness, and it, therefore, becomes an obesogenic factor in that it is tied to comfort eating and eventually to weight gain [8]. The perception that being overweight is considered good, healthy, and a sign of prosperity is accepted in South Africa and countries such as Morocco and the USA [9]; it is believed that when one has money, they can buy food and eat as much as possible, reflecting that they have enough money [7]. Another perception promoted in the African context is that mothers are encouraged to eat more for their well-being and that of their infant after childbirth, leading to excessive weight gain [9,10]. Thus, wealth and well-being are instrumental in perpetuating the increase in overweight and obesity [11]. Another common belief that drives the desire for a larger body weight remains and can be perpetuated by a stigma attached to being “thin” and the labelling of thin individuals as HIV-infected [12]; these perceptions continue to upsurge the burden of overweight and obesity among individuals and society. The food environment also makes achieving a healthy body weight challenging due to the increased availability and affordability of junk food [13]. Additionally, most South Africans are reportedly physically inactive and lead a sedentary lifestyle, the personal factors linked to obesity [9,12,14].

The increased prevalence of overweight and obesity among adults in South Africa led to the development of large-scale prevention strategies, as documented in the Strategy for Prevention and Control of Obesity document (2015–2020); with the idea of strengthening efforts to prevent and reduce the prevalence of obesity by 10% by 2020 [15]. However, the strategy failed to achieve the target set, as shown by the continual rise in the prevalence of overweight and obesity. For population-level prevention strategies to be effective, they need to be accepted and supported by the general population. Thus, tackling obesity may require other strategies, such as understanding the individual’s perception of overweight and obesity to health. Perceptions of persons who are overweight and obese may increase our understanding of how they may respond to weight reduction interventions. The use of in-depth interviews is helpful to understand better the issues of perceptions of body weight and the risks arising from obesity in a multicultural context. Thus, this study explored the perception of overweight, obesity, and health among South African adults.

## 2. Materials and Methods

### 2.1. Study Design and Setting

This descriptive qualitative study aimed to conduct in-depth interviews with adults in the three provinces of South Africa, namely, Gauteng, North West, and Mpumalanga. This study used a semi-structured interview guide to gather data from the study participants. South Africa has diverse ethnic groups: North Sotho, Zulu, Xhosa, Venda, Tsonga, Tswana, South Sotho, Swazi, and Ndebele. Thus, the provinces were chosen due to the diversity of ethnic groups residing in these settings.

### 2.2. Study Population and Sampling

Adults 18 years and older were selected to participate in the study. The study employed purposeful, nonprobability sampling where only participants who were overweight (BMI, 25 kg/m^2^–29.9 kg/m^2^) and obese (BMI, 30 kg/m^2^ and above) were selected to be part of the sample. The researchers aimed to continue interviewing participants until data saturation was achieved, whereby no new data emerged. Thus, the number of individuals volunteering to participate dictated when recruitment into the study ceased. When the recruitment ceased, there were 24 participants, which formed the study’s sample size.

### 2.3. Data Collection

A semi-structured interview schedule was used to collect data and audio-recorded interviews. Data were collected over four months, from June to November 2016. The data were collected with the use of an interview guide. The interview guide was developed in English and translated into Setswana and isiZulu, as these are the most common languages in the community. The interviews were conducted in a tent to ensure privacy. The in-depth interviews lasted for 45 min. A digital recorder was used to ensure the study’s rigour, and the data was translated verbatim. To ensure trustworthiness, the researcher read the transcripts multiple times to ensure that the understanding of the data was kept. Peer debriefing sessions between the researcher and the supervisors were held during the analysis to agree on the study’s data and findings. We made sure that all aspects of the research study were thoroughly described. Direct quotes from the respondents’ interviews were used to present the findings.

### 2.4. Data Analysis

The audio-recorded interviews were transcribed verbatim in the interview language and translated into English where necessary. To be familiar with the data, the researcher read the transcripts carefully and repeatedly. A codebook was established to keep a list of codes and definitions to allow the researchers to track how codes are used to make sense of the data. A thematic analysis was conducted using NVIVO 12 software (QSR International Inc., Burlington, MA, USA). The English versions of the twenty-four transcripts were imported into NVIVO 12 software for analysis; they were then grouped into similar concepts and contexts for interpretation. The results were then discussed in coded themes and subthemes of the topic under discussion. Finally, the themes and findings were presented through a narrative illuminated with quotes from the data.

## 3. Results

### 3.1. Respondents’ Information

Twenty-four (24) respondents participated in the study, comprising female (*n* = 18) and (*n* = 6) males. The majority of the participants, 75% (*n* = 18), were female. Sixteen (67%) of the respondents were single. The majority of participants, 75% (*n* = 18), had secondary and higher education, and 62% (*n* = 15) were unemployed. The respondents were aged between 19 years to 64 years (Table 1).

### 3.2. Themes

Four major themes emerged from the data: the perception of overweight and obesity, the perception of being overweight and obese, obesity and health, and cultural beliefs about obesity. The details of the themes are shown in Table 2 below.

#### 3.2.1. Perceptions about Causes of Overweight and Obesity

Environmental and personal factors were perceived as the leading causes of overweight and obesity and are presented below.

Eating habits

The respondents believed that obesity was caused by eating unhealthy food items such as lots of fat and starchy foods.


*“Eating a lot of fat, just like me, I eat good stuff; I just eat whatever I want” (30-year-old male).*



*“Obesity means you eat too much unhealthy food. Eating starchy foods then you go to sleep after eating” (54-year-old female).*



*“It is found that they [overweight people] eat fatty foods and they are not active” (35-year-old female).*


Type of food and drink

The respondents perceived that certain types of food and drink could cause obesity. For example, they believed that people who were overweight ate large amounts of junk food, which was unhealthy. The respondents identified sugar-sweetened beverages as the leading cause of obesity. However, they incorrectly perceived that consumption of sugar, even in large amounts, was not linked to weight gain and obesity.


*“They [overweight people] eat too much junk food. You know, people like nice food, and they are not healthy” (19-year-old female).*



*“Coke makes you gain weight, especially if you can be addicted to it and your body gets used to it, you will gain weight” (48-year-old male).*



*“No, I do not think so; sugar has its effects, especially when you eat it a lot, it can make you sick, but it does not give you weight. Sugar does not make you gain weight; it makes you have energy, but fat is what makes people gain weight” (33-year-old female).*


Lack of exercise

The respondents believed that overweight people were prone to lifestyle illnesses because they did not exercise.


*“They [overweight people] cannot exercise, so many illnesses are going to attack them because they do not exercise, they just sit and eat” (48-year-old male).*



*“These people [perwons with overweight] do not exercise; they just sit” (19-year-old female).*


Pregnancy

For women, gaining weight after pregnancy was expected and accepted. The respondents believed nothing could be done if someone gained weight because of pregnancy.


*“When I was pregnant, I gained weight, and after giving birth, I never lost it; I was still the same weight” (30-year-old female).*



*“I would not even mention when I started having children. I do not know how big I would get. That makes me angry because my belly is also a problem” (24-year-old female).*


#### 3.2.2. Perceptions about Being Overweight and Obese

Being overweight as a sign of having much money is a belief

There is speculation that men with big bellies had large amounts money (rich). However, the respondents in the study disagreed that having a big bellies was a sign of being rich. They considered people with big bellies to be eating too much of unhealthy food and had no stress in life. The respondents considered the link between money and being overweight as a cultural belief rather than a norm.


*“It is just a saying; it is the same as the one saying a person with a big stomach has a lot of money, there is no such thing, as a person with a big stomach does not have money, it is because of eating unhealthy food” (54-year-old female).*



*“What I know is that when we say a person has a lot of money when we refer to a person who is overweight because we think they do not have stress, he/she gets what he/she wants at any time, they have everything. That is why they are overweight” (22-year-old female).*


Some overweight people are healthy

One of the respondents believed that some overweight people were healthy and happy. Their body weight was attributed to being happy and having no worries in life.


*“Some [overweight people] are healthy, but mostly being overweight is caused by a happy heart if you do not have any worries and complains” (43-year-old female).*


#### 3.2.3. Obesity and Health

Risk of developing disease conditions related to obesity

It was also perceived by the study participants that there was no association between health problems and being overweight and obese.


*“There are no health problems associated with being overweight unless it is because of less weight; how are health problems associated with being overweight?” (33-year-old female).*


However, one of the respondents perceived that old age, rather than being obese led to the risk of developing disease conditions, such as diabetes and high blood pressure.


*“Yes, I can also have them [disease conditions] because I do not know the exact cause of these illnesses. Once you reach 40 years, many illnesses attack you very easily, such as high blood pressure, diabetes, and heart problems. They say the more you get old, the more illness attacks you because your body changes; they tell us that after 40 years we must expect such illness” (48-year-old male).*


The stigma attached to being thin

Thin people were labelled as being HIV-infected. Thus, individuals may choose to remain overweight due to the stigma attached to being thin. Moreover, people who were overweight and lost weight were labelled as sick and considered HIV/AIDS infected


*“It is just an old belief, but ever since this AIDS illness started, you are in big trouble when you lose weight after being overweight. They say you are sick even if it is something else than AIDS” (48-year-old male).*


#### 3.2.4. Cultural Beliefs and Obesity

Cultural beliefs on obesity

Cultural beliefs usually influence the acceptance of larger body weight among individuals, particularly in Africa. Most of the respondents in the study were from different South African ethnic groups. Although most participants perceived that overweight and obesity were acceptable in their culture, the current generation did not embrace them. Moreover, overweight and obesity among the respondents were acceptable among women.


*“In my Sotho culture, they consider obesity normal, especially in women” (22-year-old female).*



*“They [Xhosa men] want big women” (19-year-old female).*



*“In Pedi culture, they see it as a normal thing, but nowadays it is no longer a normal thing” (39-year-old female).*



*“To them [Ndebeles] it is not a good thing, having an overweight body is not good at all, Ndebeles do not want overweight bodies” (52-year-old male).*


Individual views on body weight preference

The respondents viewed body weight preference among women or men (larger or thin) as an individual choice.


*“Well, men choose, others, prefer women who are underweight, but some prefer those who are overweight.” (48-year-old male).*



*“As I said, it is a personal preference, but men I know prefer women with a fuller figure, not fat women” (22-year-old female).*



*“In terms of the preference of body weight, it goes with individual choices, we have our differences, the same applies to women, some prefer a slender man and some prefer a big man, it goes just like that” (52-year-old female).*


## 4. Discussion

The study aimed to qualitatively explore the perception of overweight, obesity and South African adults’ health and cultural beliefs. The age group consisted of younger adults who may have been living away from their parents and, therefore, not influenced by culture. Cultural beliefs are usually practiced within the family, where parents largely influence the practice. The respondents in this study believed that obesity could be caused by unhealthy habits such as eating junk food. These unhealthy eating habits are exacerbated by the increase in the availability and accessibility of junk food in African communities [13]. In the South African context, the food environment has changed such that informal, fast food outlets are everywhere, especially in urban areas, making unhealthy food easily accessible [16]. Participants in a study conducted in Nepal, pointed out that the availability of junk food was instrumental in the increase of obesity cases in that country [13].

Due to the availability of junk foods, people adopt diets that promote the consumption of high-energy-dense foods [6,12]. Therefore, the food environment becomes obesogenic as it promotes the consumption of unhealthy food items that are micronutrient deficient, leading to increased calories and weight gain [5]. In addition, increased food prices are instrumental in exacerbating overweight and obesity, as individuals in the community rely on fast foods which are cheap and readily available [12]. Increased food prices amid unemployment worsen poverty and the inability to purchase nutritious food [17]. Such findings cannot be ignored since the food consumed is less varied and has a high energy density, contributing to increased calories leading to obesity; these findings are instrumental in informing interventions that would improve the eating behaviours of individuals and society.

The respondents in the study identified the type of drinks, such as sugar-sweetened beverages, as the main culprit in causing obesity. Sugar-sweetened beverages are linked to obesity due to increased energy intake and frequent consumption, leading to weight gain [18]. In a study conducted among Mexican adults [19], it was observed that increased sugar-sweetened beverages were associated with weight gain. Furthermore, Kim et al. [20] also found that higher sugar-sweetened beverages were linked to an increased risk of obesity. A higher intake of sugar-sweetened beverages has been reported among adults in South Africa [21]. However, the current study did not measure the frequency of sugar-sweetened beverage consumption. The increased consumption of sugar-sweetened beverages tends to be obesogenic because it is associated with a high total body fat percentage resulting in weight gain [22]. Even though the sugar tax has been adopted and implemented in South Africa, adverts for sugar-sweetened beverages in the media and billboards undermine efforts to address and reduce obesity, especially among individuals who do not see the consumption of higher amounts of sugar as a problem.

The respondents further believed that people who were overweight were prone to weight gain due to a lack of exercise. The most striking observation from the study is that respondents self-impose their beliefs to be those of other respondents other than themselves. However, we should take cognizance of the fact that the current study did not assess if the respondents engaged in exercises or not. Okop et al. [9] reported that only a few respondents engaged in any form of physical activity in a study conducted among Ghanaian adults. Reportedly, adequate exercise is vital in the management of obesity [11]. Studies have shown that most individuals lived a sedentary lifestyle and were at risk of becoming obese [8]. Similarly, Motadi et al. [5], in a study conducted in Limpopo, reported that most females were inactive, exacerbating overweight and obesity among individuals and impairing weight management strategies.

The study’s findings showed that women perceived that they were prone to being overweight due to pregnancy and childbirth. This is likely because women who had just given birth tended to increase food consumption [13]. Similar sentiments were echoed by Agyapong et al. [11], who reported that women of childbearing gained weight during and after childbirth since they are expected to eat more in order to promote the baby’s well-being. Thus, the practice of increased food consumption after giving birth, promoted in most African cultures, leads to excessive weight gain and leads to normalizing this weight gain. Thus, for obesity prevention intervention strategies to work, women must be educated on quality diet intake during and after pregnancy.

As reported in the study, the respondents believed that older men with pot bellies (big stomachs) were wealthy. These findings are similar to those reported in a study conducted in Ghana, where overweight, in general, was linked to wealth [9]. These beliefs indirectly support the notion that overweight and obesity were acceptable and could, therefore, hinder the intervention strategies that are put in place to combat overweight and obesity. The findings, in essence, demonstrate that individuals in communities influence intervention strategies’ implementation through their cultural beliefs.

Some respondents also believed that overweight individuals were healthy and that their weight was attributable to their happiness in life. Moreover, respondents also believed that poor people were inclined to gain weight as long as they were happy. This cohort believed that happiness was obesogenic and thus, led to weight gain [6]. According to a study conducted in KwaZulu-Natal, South Africa, such beliefs are responsible for increasing the prevalence of overweight and obesity in communities. The implication is that individuals who are obese and perceive themselves to be healthy and happy may not take any action to improve their health.

Although some respondents believed that they were at risk of developing disease conditions like diabetes and high blood pressure, they attributed the risk to ageing rather than obesity. Reportedly, as individuals age, excessive weight comes with health risks like diabetes and cardiovascular diseases [23]. In this study findings, participants perceived that growing older made the health risks related to being overweight and obese unavoidable. However, it is essential to note that this was reported by younger female respondents, which was consistent with findings from Jackson et al. [24], showing that as individuals aged, they gained weight. As little is known about older adults’ perception of their body weight in South Africa, understanding the perception of aging and the health risks associated with being overweight and obese are crucial for implementing appropriate strategies to combat overweight and obesity in South Africa and elsewhere.

The findings also showed that individuals might remain obese to avoid the stigma of being thin. This is so because respondents are aware of overweight individuals who lost weight, were labeled as sick, and considered HIV/AIDS infected. As evidenced by studies done in communities where there was a high prevalence of HIV, HIV-infected individuals preferred to have high body weight to avoid the stigma attached to thinness [8,16]. Such stigma exacerbates the burden of overweight and obesity. The stigma is also detrimental to implementation of obesity management strategies.

In the current study, respondents believed that obesity was acceptable in their culture, although they did not embrace the belief themselves. This was likely because the participants were young to middle age and had a level of education that informed them about the benefits of losing weight. In South African studies, culture is still primarily linked to accepting overweight and obesity [25]. For example, earlier South African studies consistently reported that black men preferred larger women [26,27]. Subsequently, studies done in the U.K based their assumption that individuals who were black preferred larger body weight [28,29]. The current study’s findings have de-mystified these beliefs by showing that individuals will have different preferences, not necessarily based on cultural norms. These results have tremendous implications for public health in general, as it is clear that overweight and obese individuals may no longer hide behind culture.

## 5. Study Limitations

This was a qualitative study, and the findings cannot be generalized as perceptions differ from individual to individual. Another limitation is that people who were not overweight were not asked if they thought they could develop overweight and obesity. Future studies should include those who are not overweight to expand our findings.

## 6. Conclusions

This study has shown that participants are aware that unhealthy eating habits contribute to weight gain, subsequently leading to overweight and obesity. The findings further showed that body weight preference was personal and differed from individual to individual. The study provided insight into perceptions of obesity from individuals’ perspectives. The misperceptions of overweight and obesity are helpful in the design of impactful intervention programmes that can be used to prevent and manage obesity in South Africa.

## Figures and Tables

**Table 1 ijerph-19-12335-t001:** Sociodemographic characteristics of the participants (*n* = 24).

Sociodemographic Characteristics	*n*	%
Age		
19–25 years	3	13
26–35 years	8	33
36–45 years	5	21
46 years and above	8	33
Gender		
Male	6	25
Female	18	75
Ethnic group		
Ndebele	4	17
Zulu	2	8
Northern Sotho (Pedi)	8	33
Tsonga	3	13
Tswana	4	17
Venda	3	12
Marital status		
Single	16	67
Ever married	8	33
Education level		
No primary education	2	8
Primary education	4	17
Secondary education and higher	18	75
Employment status		
Employed	9	38
Unemployed	15	62

**Table 2 ijerph-19-12335-t002:** Themes and sub-themes.

Themes	Sub-Themes
Perceptions about the causes of overweight and obesity	Eating habits Type of food and drink Lack of exercise Pregnancy
Perceptions about being overweight and obese	Being overweight as a sign of having much money is a belief. Some overweight people are healthy
Obesity and health	Risk of developing disease conditions related to obesity Stigma attached to thin
Cultural beliefs and obesity	Cultural beliefs on obesity Individual views on body weight preference

## Data Availability

The data presented in this study are available upon request from the corresponding author.

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
