# Peer review of "The Perception of Overweight and Obesity among South African Adults: Implications for Intervention Strategies"

_ijerph, 2022, doi:10.3390/ijerph191912335_

Round 1
Reviewer 1 Report
Dear Authors:
The article is interesting but some aspects need to be improved:
1) I suggest that when you referring "obese people or obese person" in te text, please change it to "person who living with obesity or person (people) with obesity". Person-centered language should be used.
2) Line 35: I suggest to change "metabolic abnormalities", maybe you should say metabolic comorbidities.
3) Paragraph between lines 63-69: the idea is a bit confusing, which author are you referring to?
4) I suggest to the author check the language with a native English person.
Reviewer 2 Report
The manuscript entitled “The perception of overweight and obesity, health, and cultural beliefs among South African adults: implications for intervention strategies” explores perceptions and beliefs about being overweight or obese among South African adult individuals. Despite having its merit, the manuscript contains several limitations that authors should address point-by-point before publication endorsement.
Introduction
- The authors should provide the rationale behind using in-depth interviews as the primary tool to examine the perception of overweight and obesity among the study subjects.
-The authors should shorten the introduction section because it is repetitive and too long.
Material and Methods
-Please provide the criteria used to define overweight and obesity in the study subjects (lines 92-93).
-Did the authors calculate the sample size? In my viewpoint, an n = 24 appears insufficient when analyzing interview information.
-Information regarding the statistical approach to analyze and compare data from in-depth interviews is missing.
Results
- The sample size is too small, which does not allow for analyzing the potential effects of age, gender, marital status, and education level, among others, on the perception of overweight and obesity. So, please adjust data from Table 2 by confounding variables such as those mentioned above.
- I encourage the authors to examine data from interviews by multivariate analysis to describe what factors (i.e., age, gender, marital status, education level, etc.) influence the perception of being overweight or obese.
-Interview fragments should be removed from the result section and provided as supplementary files. Reading the main findings turned hard due to interview fragments in their current form.
Discussion
- Discussion is too long. Please shorten it. Also, compare your findings with data from other research teams, even if contradictory.
- Please discuss the advantages of using in-depth interviews instead of other methodologies to collect data regarding the perception of being overweight or obese among the study subjects.
-Include limitations to the study.
Round 2
Reviewer 2 Report
I kindly request the authors to submit the review report point-by-point. In its current form, I cannot find the correspondence of changes to the main manuscript text with each point of the review report. So, please resubmit the review report, describing page and line numbers where I can see every change made to the manuscript.
Thank you.
